# Artefact Retrieval: Overview of NLP Models
# with Knowledge Base Access

**Vilém Zouhar**                                    VZOUHAR@LSV.UNI-SAARLAND.DE
**Marius Mosbach**                                MMOSBACH@LSV.UNI-SAARLAND.DE
**Debanjali Biswas**                                DBISWAS@COLI.UNI-SAARLAND.DE
**Dietrich Klakow**                                  DKLAKOW@LSV.UNI-SAARLAND.DE
*Saarland University, Saarland Informatics Campus, C7 1, R. 001*
*66123 Saarbrücken, Germany*

## Abstract

Many NLP models gain performance by having access to a knowledge base. A lot of research has been devoted to devising and improving the way the knowledge base is accessed and incorporated into the model, resulting in a number of mechanisms and pipelines. Despite the diversity of proposed mechanisms, there are patterns in the designs of such systems. In this paper, we systematically describe the typology of *artefacts* (items retrieved from a knowledge base), retrieval mechanisms and the way these artefacts are *fused* into the model. This further allows us to uncover combinations of design decisions that had not yet been tried. Most of the focus is given to language models, though we also show how question answering, fact-checking and knowledgable dialogue models fit into this system as well. Having an abstract model which can describe the architecture of specific models also helps with transferring these architectures between multiple NLP tasks.

## 1. Introduction

For multiple NLP tasks and primarily language modelling, the BERT [Devlin et al., 2019], GPT-2 [Radford et al., 2019], GPT-3 [Brown et al., 2020] and T5 [Raffel et al., 2020] models have seen great success. Their performance is, however, limited on rare or unseen entities [Logan et al., 2019, Schick and Schütze, 2020] and knowledge-intensive NLP tasks [Chen, 2020]. As a consequence, they perform poorly on the task of fact-aware language modelling, where the words to be predicted are named entities [Logan et al., 2019].

Traditionally question answering systems relied heavily on knowledge base access. This has been expanded by other NLP tasks for which models have been proposed (Appendix A) that make use of knowledge bases. Not only do they perform better on e.g. fact-aware language modelling, but they can also provide a degree of explainability (which fact was retrieved) and allow for the model and knowledge base components to be trained separately. The latter is a strong prerequisite for efficient knowledge base manipulation and control, such as removing misinformation or biases [Bhardwaj et al., 2020, Metz, 2019] or adding new information. This is problematic for models like BERT or T5 because their knowledge is tightly coupled with their generative ability, which is stored implicitly in the parameters. Such models make use of parametric knowledge representations as opposed to non-parametric ones, which are typically used in retrieval-based approaches (separation of the memory and generation components). Separating memory from the generative component allows for more flexible architectures, though at the cost of increased complexity.

While using knowledge bases for language modelling is not immediately intuitive, it is very prevalent in the area of open-domain question answering. Commonly in this case the retrieved

knowledge (passed to a generative model) is useful for producing an answer to a question in natural language as the model is trained to condition its generation on the retrieved artefact. Having access to a knowledge base is mandatory for the task of extractive question answering where the output is a span from the available data. Pre-trained language models have also been shown to be able to perform question-answering in the form of being primed by the question [Liu et al., 2019]. This demonstrates the amount of knowledge that is stored in their parameters, though these models are usually outperformed by models with explicit knowledge base access [Petroni et al., 2020]. Models in the task of slot-filling and fact-checking also benefit from explicit memory access largely because of the same reasons.

Recently there has been an effort in the community to phrase any NLP task in a unified fashion as a sequence to sequence task (seq-2-seq). Examples of this are phrasing multiple tasks as question answering [McCann et al., 2018], the T5 model [Raffel et al., 2020], the focus on prompt priming [Le Scao and Rush, 2021, Lin et al., 2021] and GPT3 and its few-shot learning approach for multiple tasks [Brown et al., 2020]. There is also a separate line of research that aims to inject commonsense or factual knowledge into language models by training them on data derived from knowledge graphs. An example of this is injecting numerical reasoning skills [Geva et al., 2020] or commonsense knowledge [Bosselut et al., 2019]. This is orthogonal to retrieval-based approaches (such as dialogue generation with commonsense knowledge base access by Young et al. [2018]) on which we focus in this paper.

**Contribution.** The goal of this paper is to provide a formalism for an abstract model which underlies many specific models for knowledge-intensive tasks with knowledge base access. Similarly to KILT [Petroni et al., 2020], we hope that this provides a foundation for future research into task-agnostic memory and model architectures. Lastly, this systematic approach to model description allows us to uncover (1) combinations of mechanisms tried only on one task, which could, however, also work on other NLP tasks and (2) combinations of mechanisms that had not yet been explored for any task.

**Outline.** In Section 3 we introduce the abstract model underlying other systems together with the different components and the way they can be instantiated. The limitations of this schema and also future work with methods not yet explored are discussed in Section 4. We conclude in Section 5.

An important part is also Appendix A in which different models and approaches for language modelling, question answering, fact-checking and knowledgable dialogue are examined to show how disparate models fit into this schema.

## 2. Related Work

A comprehensive overview of earlier work on neural model-based information retrieval systems together with a general introduction has been done by Mitra and Craswell [2017]. More recently models such as BERT have been utilized successfully for the task of retrieval itself [Nogueira et al., 2019b, Soleimani et al., 2020].

The aim of KILT [Petroni et al., 2020] is to provide a common knowledge base for a number of different NLP tasks (ranging from question answering to fact verification) and to stimulate research in task-agnostic memory architectures. Reformulating various NLP tasks to all use the same knowledge base format provides a stepping stone for the formalisms of artefact retrieval.

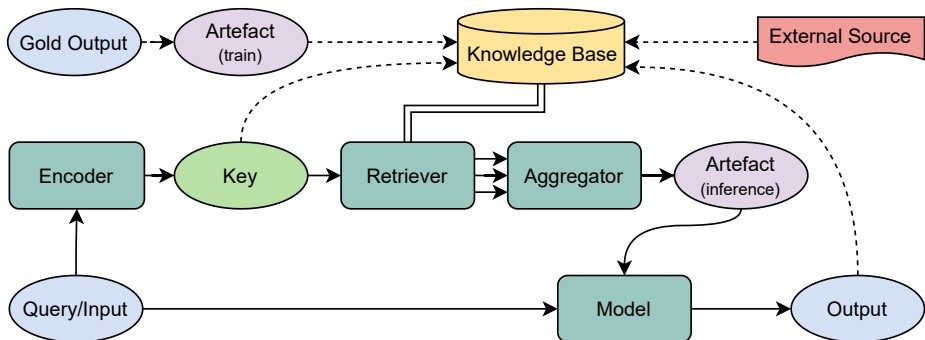

Figure 1: General scheme of NLP models utilizing artefacts by retrieving them from a knowledge base and fusing them into the model in order to produce a better output. Dashed links are utilized only in knowledge base creation and usually not all at once.

Defining a task-agnostic abstract model is closely related to multi-task learning. The goal of this approach is to improve the performance by training the model on multiple tasks rather than on individual ones [Maillard et al., 2021]. The hope is that representations and generalizations learned for one task will help on another one and vice versa. A strong requirement for this is that the instantiations for different tasks (in the multi-task setup) share significant portions of the model.

An edge-case of this is using a pre-trained BERT model and then fine-tuning it for the new task and/or possibly adding extra layers to match the input and output shapes. Even for BERT, however, it was shown several times [Petroni et al., 2020, Liu et al., 2019, Sun et al., 2019a, Kim et al., 2019] that training on multiple tasks improves the performance [Aghajanyan et al., 2021]. This would not be possible without a common model shared among the tasks. Further related work is discussed in the respective sections when presenting individual NLP models and how they fit into this schema.

## 3. Artefact Retrieval

Systems utilizing knowledge bases usually contain up to four main components (excluding the knowledge base itself): **encoder**, **retriever**, **aggregator** and **model**. In other works, some of these parts are joined together, most commonly the encoder, the retriever and the aggregator. We separate them for clarity.

A *query* or an *input* is specific to the given task. For language modelling, it is the previous context, for question answering the question, for slot-filling usually the entity and the relation, and for fact-checking the fact to be verified. In the context of this work, a *knowledge base* is a collection of items, usually (but not necessarily) with a pre-built index that maps keys to values. Prototypically, it is a collection of *documents*, though it can also be a collection of gold training data input-output pairs or a knowledge graph. *Candidates* are values retrieved from the knowledge base, which may be later post-processed (e.g. reranking or averaging) by the aggregator to form an artefact. *key* is an object through which the retriever finds suitable artefacts. Commonly the key is a dense vector representation of the input, though it is not necessarily a vector and may be dependent also on an intermediate model computation. An *artefact* is an object which is (1) dependent on elements retrieved from the knowledge base (e.g. the concatenation of $k$ retrieved documents) and (2) can be

used to improve the performance during training and inference. In the simplest example, it is the retrieved value itself, though it can also be multiple retrieved values or their combination.

On a simplified level, models with artefact retrieval usually work the following way, given a query/input $q$ and knowledge base $\mathcal{B}$:

$$
\begin{aligned}
\text{[Key] } k &= \texttt{Encoder}(q) \\
\text{[Candidates] } C &= \texttt{Retriever}(\mathcal{B}, k) \\
\text{[Artefact] } \xi &= \texttt{Aggregator}(C) \\
\text{[Output] } \hat{y} &= \texttt{Model}(q, \xi)
\end{aligned}
$$

A diagram of this pipeline can be seen in Figure 1 (ignore dashed connectors). Pipelines without knowledge base access would only make use of the bottom line (Query/Input → Model → Output). The aggregator is shown to only depend on the retriever output but in some scenarios may also take the key or the query itself as an input.

## 3.1 Artefact Typology

Most systems used in the literature differ in the encoder, retriever, and model design. We bring attention to four properties that characterize the differences between such systems.

- **Fusion** (early, late, other)
- **Specificity** (sample, task, class)
- **KB source** (train, external, dynamic)
- **Key & value type** (dense, sparse, other)

## 3.2 Fusion

Formally the model estimates $p(y|x, \xi)$ where $y$ is the ground-truth output, $x$ is the query/input and $\xi$ is a retrieved artefact. Fusion Sun et al. [2018] concerns with which point is the artefact made available to the model. It can be presented to the model at the same time as the query/input, e.g. by concatenating $x$ and $\xi$ (early fusion), just before the output is created by the model (late fusion), or somewhere in between. Formally, the model computation is a composition of functions $f_1, \ldots, f_n$. In the simplest example of feed-forward networks, these correspond to single layers and activation functions and on a higher level, they correspond to whole encoder/decoder blocks. The distinction as to what counts as early and late is not clear and for presentation purposes, we consider early fusion at the level of $f_1$ and late fusion at the last stage, $f_n$. These functions themselves may, however, still be composed of multiple others. In **early fusion**, the artefact is the input together with the query to the first function $f_1$, while in **late fusion** the query is the single input to $f_1$ and artefact is considered only for $f_n$.

$$
\begin{aligned}
\text{No fusion:} \qquad & f_n \circ \ldots \circ f_2 \circ f_1(q) \\
\text{Early fusion:} \qquad & f_n \circ \ldots \circ f_2 \circ f_1(q, \xi) \\
\text{Late fusion:} \qquad & f_n(f_{n-1} \circ \ldots \circ f_1(q), \xi) \\
\text{Intermediate fusion:} \qquad & f_n \circ \ldots \circ f_k(f_{k-1} \circ \ldots \circ f_1(q), \xi)
\end{aligned}
$$

Intuitively it makes more sense to prefer early fusion, to maximize the model's access to extra information [Izacard and Grave, 2020, Karpukhin et al., 2020]. However, this can also be a disadvantage, as the signal from the artefact can get lost during the long computation. In the case of an artefact which is the gold output of a similar query from the training data, later fusion makes more sense. This also allows for a degree of explainability. By examining the forward pass of the last function we could determine what the contribution of the artefact was to the produced output.

The decision of how late the fusion should be depends heavily on the artefact type. The application of every function in the chain of computation projects the input to some latent space. The final function $f_n$ is special because it projects the output of previous functions to the space of possible outputs for the whole model. In this space, there is the prediction $\hat{y}$ and also the true output $y$. The task performance metric is defined in this space. During inference, adding an artefact should ideally move the prediction in the output space closer to the correct output. Assuming $c$ is the intermediate computation and there are two (overloaded) functions that produce a prediction: $\hat{y}_x = f_n(c)$ and $\hat{y}_\xi = f_n(c, \xi)$. In circumstances in which adding the artefact helps, $L(\hat{y}_\xi, y) < L(\hat{y}_x, y)$, where $L$ is a loss function such as cross-entropy. This is illustrated in the first row of Figure 2 for $n = 2$.

Assume that we can create an inverse of the last projection and see where the correct output lies in the intermediate representation. There may be multiple such $c_t : f_n(c_t) = y$ or none, if too much information was lost by the first projection $f_1$. Further, assume that there is always at least one such $c_t$. We may then define an intermediate loss $L_i$ for each model computation by measuring the distance of the partial computations to the back-projection. Similarly to late fusion, we consider two overloaded functions that produce the intermediate representation $c_x = f_1(q)$ and $c_\xi = f_1(q, \xi)$. Adding the artefact then ideally moves the intermediate representation closer to the back-projection and reduces the intermediate loss: $L^i(c_\xi, c_t) < L^i(c_x, c_t)$.[1]

This is illustrated in the second row of Figure 2, which depicts a model with only two computational steps: $f_2 \circ f_1$. Early fusion (second row) adds the artefact to $f_1$, while late fusion adds it in the next step. For simplicity in the figure, we consider the standard $L^2$ distance loss between the points. In both cases, adding the artefact reduced the target loss. For early function, the intermediate loss was also reduced and the target loss was lower. This does not always happen and complex computations may still at some point project the intermediate computation to the same point regardless of whether an artefact was added earlier or not. It may also be the case that training with artefacts takes a longer time and the intermediate loss is higher but that the presence of the artefacts will make the model converge to a better optimum (lower generalization error).

Even though the invertibility of projections in this paper is only used as an illustration for the artefact fusion and an intermediate loss does not have to be defined in practice, invertible neural networks are an ongoing topic of research [Ardizzone et al., 2018, Behrmann et al., 2021]. The combination of artefact retrieval and invertible neural networks has not yet been explored to our knowledge.

**Fusion Mechanisms.** There are common patterns with respect to artefact fusion. *Priming*, as used by many question-answering systems [Guu et al., 2020, Lewis et al., 2020, Karpukhin et al., 2020], is an example of an early fusion technique. In the simplest scenario, the retrieved paragraphs are prepended to the actual question and put together to the model as one input.

---

1. In case of multiple elements that map to $y$, we can define a loss that considers the minimum distance to any of them: $L^{i'} = \min L^i(c, c_t)$. If there is no such element in the projection space, then we may consider the elements that project close to the target $C_t = \arg \min L(f_2(c), y)$.

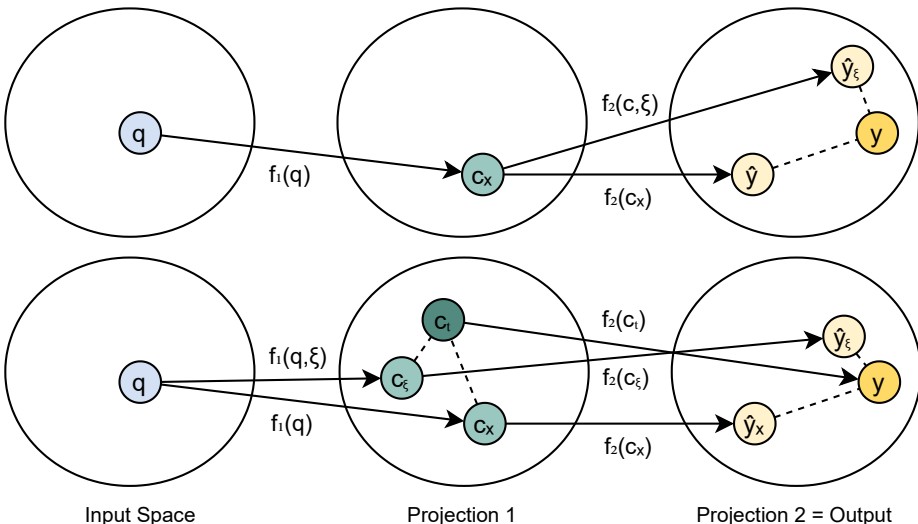

Figure 2: Example of how late (top) and early (bottom) fusions affect the projections into interme-
diate spaces. The lengths of dashed lines correspond to the loss (longer is greater loss).
Adding an artefact $\xi$ to the computation decreases the loss in both early and late fusions.

An extension to this is *key-aware priming*, in which the keys of the retrieved values are also
put to the model in an early fusion. The motivation for this in the context of question answering
is that the keys may be encoded questions and the retrieved value is simply the answer. Then at
inference time, the system can receive a specific question of which the *negation* is stored in the
knowledge base. Knowing the answer to the negation of a question may be beneficial in answering
the original question, though knowing the question to which the answer corresponds is vital. Key-
aware priming can be also conceptualized as simply storing the key-value pairs as: (key, key+value)
and then retrieving the second item of the tuple by the retriever which would also retrieve the key.

For multi-class predictors[2] like language models it is possible to consider the training data as
a knowledge base and when retrieving, suggest k nearest neighbours. This can be followed up by
averaging the probabilities of the output classes based on the neighbour distance (performed by the
aggregator). For this late fusion approach, *output gating*, the last function $f_n$ can either be a simple
convex combination $\hat{y} = \lambda \cdot c + (1 - \lambda) \cdot \xi$ or a slightly more complex gating mechanism, which
computes $\lambda$ dynamically based on the current input [Khandelwal et al., 2019, Yogatama et al., 2021].
Both are discussed in detail in Appendix A.1.

Filtration/masking can be seen as very late fusion. For slot-filling or fact-aware language mod-
elling, the retriever may provide relevant documents out of which a term incidence vector[3] is con-
structed. The output of the model is then masked and normalized to prevent outputs unrelated to the
query ($\hat{y} = \frac{c \odot \xi}{|c \odot \xi|}$). An example of this is to constrain beam-search decoding to valid names only as
done in the recently proposed GENRE model for entity retrieval tasks [De Cao et al., 2020].

---

2. Single-token predictors can be treated as multi-class predictors with the classes being the vocabulary.
3. A vector $v$ of the length of the vocabulary filled with ones and zeroes. Then $v_i = 1$ if the $i$-th term occurred in the
retrieved documents, otherwise zero.

### 3.3 Specificity

The property of specificity is not directly bound to the pipeline of models using artefact retrieval. In a wider sense of an artefact being something that improves inference-time performance, the retriever can produce artefacts that help the model, though which are not necessarily dependent on a knowledge base. In this specific case, the retriever can simply ignore the conditioning on the knowledge base and can be directly fused together with the encoder. In the usual scenario, the encoder makes decisions based on the current query. It may, however, ignore significant parts of the query or be conditioned on a specific task and not take the query into consideration at all.

For the task of open-domain question-answering, retrieving documents based on their embedding's inner product similarity to the query embedding [Karpukhin et al., 2020] is an example of the sample-specific property. Presumably significant portions of the query are considered when producing the key for the retriever and different queries produce different artefacts (sets of documents).

Pre-trained language models, such as BERT or GPT-3, are either fine-tuned on a new task or are primed on prompts either crafted by humans [Haley, 2020, Misra et al., 2020, Petroni et al., 2019] or found automatically [Shin et al., 2020, Jiang et al., 2020]. These prompts (also called stimuli, constraints, or demonstrations) are then dependent only on the specific task at hand and nothing else. As an example, Radford et al. [2019] affix the prompt `TL;DR` after a passage to induce summarization behaviour. Crafting task-specific artefact, specifically prompts, is known as *prompt programming* or *prompt engineering* [Reynolds and McDonell, 2021] and allows the models to perform a wide variety of NLP tasks.

The prompts may be in a structured form with only some parts of it being considered by the encoder. In the case of fact retrieval [Shin et al., 2020], the input is in the form of (subject, relation). It is then possible to have different prompts based only on the relation (class). Different conditioning which results in distinct encoder specificities are summarized in the following equations:

$$\text{Query/input } q = \{q_1, q_2, \ldots q_n\}$$

Sample-specific: $\qquad\qquad \texttt{Encoder}(q, \text{task})$

Task-specific: $\qquad\qquad \texttt{Encoder}(\text{task})$

Class-specific: $\qquad\qquad \texttt{Encoder}(q' \subset q, \text{task})$

### 3.4 Knowledge Base Source

The way the knowledge base is created is tightly coupled with its fusion purpose. There are, however, three common underlying patterns that describe the possible creation schemes. In the general architecture overview of pipelines with artefact retrieval in Figure 1, they are depicted with dashed lines. The most common example is in the task of open-domain question-answering, where the knowledge base, a collection of documents, is a vital component to produce the answer. A collection of documents or knowledge graph structures are examples of knowledge bases populated externally.

A specific contrast to this is having access to the training data during inference. Here, the knowledge base is created as:

$$\mathcal{B} = \{(\ \texttt{Encoder}_{\text{query}}(q), y) | (q, y) \in D_{\text{train}}\}$$

At inference time, the model can then refer back to examples it has already seen with the retriever providing e.g. a weighted combination of nearest neighbours [Khandelwal et al., 2019] or

any other aggregation of found samples. This memorization based approach is usually used with late fusion because, at the end of the computation, the model has access to its proposed output and also to the gold outputs of similar queries. These can be then combined together using e.g. gating.

Finally, the knowledge base can also be created dynamically during inference, such as by keeping a track of already predicted words in language modelling. This knowledge base creation is depicted by the dashed line from *Output* to *Knowledge Base* in Figure 1. This usage of knowledge base lies, however, slightly out of the traditional meaning and understanding of the term.

### 3.5 Key & Value Type

Commonly the knowledge base is either (1) a collection of candidates (e.g. documents, paragraphs or gold outputs from the training dataset), in which case they are retrieved by similarity to the query or (2) a more structured source of information, such as a knowledge graph.

**Vector Space Model.** Traditionally, question-answering and other NLP systems would use a vector space model for the values (documents) and queries. Such representations usually describe the algorithm for computing the vector representations for both the document and the query. These include standard statistical methods such as TF-IDF [Salton and McGill, 1986] or BM25 [Robertson et al., 1995]. Their disadvantage is that the resulting vectors have the dimension the same as the number of words in the vocabulary. A contrast to this is dense vector representation. LSA [Dumais, 2004] is one of the oldest of such methods, but it also includes learned embedding methods based on word2vec [Mikolov et al., 2013], doc2vec [Le and Mikolov, 2014], docT5query [Nogueira et al., 2019a], CLSM [Shen et al., 2014] or BERT [Devlin et al., 2019]. The latter allows for the use of the gradient signal of the latent variable (documents) to fine-tune the index, as done asynchronously by Guu et al. [2020].

Splitting the retrieval mechanism into an `Encoder` and `Retriever` is not strictly necessary and only follows a common pattern found in many systems. This is usually done for training and speed purposes because e.g. recomputing embeddings for all documents bears too high of a cost for just single retrieval. In the vector space model, an index (keys for the documents) would be built usually in the pre-processing phase and the knowledge base would constitute a mapping from this index to the documents. Then a vector similarity, usually the cosine similarity, is used which results in the following pipeline:

$$
\begin{aligned}
\mathcal{B} &= \{(\texttt{Encoder}_{\text{doc}}(d), d) | d \in C\} \\
k &= \texttt{Encoder}_{\text{query}}(q) \\
\xi &= \texttt{Retriever}(B, k) = \arg \max_{(v,d) \in B} \text{sim}(k, v)
\end{aligned}
$$

Usually, instead of retrieving just the $\arg \max$, the top-k scoring documents are returned. When using cosine similarity with normalized vectors, we may substitute the similarity by the inner product $k \odot v$, which can be approximated efficiently by Maximum Inner Product Search (Approximate Nearest Neighbour Search) algorithms in sublinear query time [Johnson et al., 2019, Guo et al., 2020, Yang et al., 2021]. Instead of documents, which are replaced by paragraphs or spans of texts, depending on the specific work. It is also possible to store query representation from the training data, as described in Section 3.4. The setup is then similar as for document retrieval, with the exception that instead of documents, the retrieved values are gold outputs from the training dataset. The inner product search remains the same.

**Knowledge Graphs.** A vastly different approach has to be chosen when the knowledge base is more structured, such as a knowledge graph. In this case, the knowledge base is usually a directed labelled graph: a set of triples (parent, relation, entity). The parent and the entity are elements of a fixed set of entities and the relation is an element of a fixed set of possible relations (usually orders of magnitudes smaller than the set of entities).

Question-answering over knowledge graphs is a specific, vastly explored [Bao et al., 2016, Lukovnikov et al., 2017] subfield of non-extractive open-domain question answering. Popular knowledge graphs, based on Wikipedia, include DBpedia [Auer et al., 2007], Wikidata [Vrandečić and Krötzsch, 2014] and YAGO [Rebele et al., 2016]. A comparison of them has been composed by Ringler and Paulheim [2017]. Specific datasets for testing question answering over knowledge graphs are WebQuestions [Berant et al., 2013] and SimpleQuestions [Bordes et al., 2015]. Knowledge graphs are, however, also used for fact-aware language modelling [Logan et al., 2019] or fact-checking [Ciampaglia et al., 2015, Tchechmedjiev et al., 2019]. Simple slot-filling without any reasoning (either multi-source facts or resolving aliases) would be a trivial task. It is then used in the opposite direction, for automatically creating knowledge graphs [Yu et al., 2014]. These knowledge bases found their use even in more distant tasks, such as Word Sense Disambiguation [Bevilacqua and Navigli, 2020]. Knowledge graph retrieval is then reduced to finding variables given constraints (a subgraph with free variables that needs to be matched over the knowledge base).

The query construction (which entity and relation should be selected) is handled by the `Encoder` and is commonly limited to a single restriction (single triplet). The `Retriever` is built on top of a database, which stores the knowledge graph. This is vastly faster than MIPS and computing the index in vector space models, but at the cost of a more complicated encoder and constraints to the type of knowledge stored.

## 4. Discussion

Descriptions of specific systems and approaches for language modelling, knowledge graph language modelling, question answering, fact-checking and knowledgable dialogue in the paradigm of artefact retrieval can be found in Appendix A. Summary characteristics of the different models are shown in Table 1.

### 4.1 Limitations

Even though the abstract model presented in this paper is versatile in the description of various models, it may be lacking in subtle ways to describe more complex systems. This complexity may be included in one of the components in order to make it fit, though this obscures clear understanding and comparison across models. An example of this is the implicit aggregation in the form of re-ranking as the final step of the DPR question answering system [Karpukhin et al., 2020]. We include it in the core model component, though a better understanding of this model would be achieved by adding an additional aggregator element to the abstract model. This would, however, conflict with most other systems which would not make use of this and for full description, this part of the pipeline would have to be set to simply identity or NOP. This has happened with other models, as documented in Table 1.

| Model | Fusion | KB Source | Keys | Values | Aggregation |
|---|---|---|---|---|---|
| k-NN LM [Khandelwal et al., 2019] | Very late Static convex combination | Train-time | Prefix embd., $L^2$ | Target word | Softmax |
| Continuous Cache LM [Grave et al., 2016] | Very late Static convex combination | Dynamic | Prefix embd., inner product | Target word | Softmax |
| Dynamic Gating LM [Yogatama et al., 2021] | Late Dyn. convex combination | Train-time | Prefix encoding, inner product | Target word | Softmax sum |
| Knowledge Graph LM [Logan et al., 2019] | Intermediate Constraints | External | Entity+relation Discrete struct. | Matching entity | None |
| Dense Passage Retrieval [Karpukhin et al., 2020] | Early Input | External | Passage embd., inner product | Passages | None |
| Nearest Neighbour QA [Lewis et al., 2021] | No model | Train-time | Passage embd., inner product | Answers | None |
| CBR-KBQA [Das et al., 2021] | Query creation | Train-time External | Query embd., inner product | Logical forms | New query |
| PullNet [Sun et al., 2019b] | Subgraph creation | Multiple External | Entities | Docs and Facts | Iterative join |
| Universal Schema QA [Das et al., 2017] | Intermediate Retrieval | Multiple External | Query embd. Attention | Facts | Iterative projection |
| FAKTA [Nadeem et al., 2019] | Early Input | External Online | Condensed query | Docs | Re-ranking, Filtering |
| Wizards of Wikipedia [Dinan et al., 2018] | Intermediate Addition | External | Context+topic, inverted index | Passages | Attention (topic) |

Table 1: Categorization of described NLP systems in terms of the artefact retrieval typology. *Fusion* describe both where it occurs and what mechanism it employs, *Keys* describes not only the key type but also the retrieval mechanism (e.g. metric).

## 4.2 Future work

An immediate observation from Table 1 is that models for knowledge-intensive tasks, e.g. question answering, do not tend to utilize late fusion nor training-time knowledge base sources yet. In the opposite directions, experiments could be made with early fusion for language models. They could also make greater use of knowledge bases with external sources.

Invertible neural networks should be studied in the context of artefact retrieval to determine the exact properties of different fusion mechanisms (e.g. quantifying the discussion in Section 3.2).

Finally, the area of using multiple separate artefact retrieval pipelines is currently unexplored. This would mean utilizing either (1) multiple knowledge bases with retrieval systems based on the same principle, e.g. dense vector representations or (2) a vastly different knowledge bases with separate retrieval systems, e.g. one with dense vector representation and the other with a knowledge graph.

## 5. Summary

In this paper, we presented an abstract model which can describe various systems for NLP tasks utilizing knowledge bases. The pipeline consists of multiple components: encoder, retriever, aggregator and the core model, typically a generative model, itself. The abstract model description leads to several key characteristics: fusion, specificity, knowledge base source and key & value type which share important properties across approaches to model design.

In the Appendix, we showed how several increasingly complex language modelling models proposed in recent years can be described in this paradigm. This is followed by model descriptions in the context of question answering, fact-checking and knowledgeable dialogue.

## Acknowledgments

This work was funded by the Deutsche Forschungsgemeinschaft (DFG, German Research Foundation) – Project-ID 232722074 – SFB 1102.

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

## Appendix A. System Description

This section provides a brief survey of different NLP systems and how their retrieval and fusion mechanisms work in the paradigm of artefact retrieval. Summaries with respect to this typology are shown in Table 1.

### A.1 Language Modelling

The common goal of language modelling is to predict the next word (distribution) given the context of the previous tokens [Bahl et al., 1983]. The performance is measured by perplexity: the inverse of the geometric average of the whole sequence probability (lower is better). The input to the model is the text prefix and the output of the following word. The prefix is usually the object that is passed to the encoder.

**k-Nearest Neighbours.** The language model proposed by Khandelwal et al. [2019] utilizes memorization of the training data to decrease the perplexity. In this approach, the authors first compute representations of all sentence prefixes (keys) and store them together with the following word (values). They then use this for the next word prediction by softmaxing negative $L^2$ distances[4] of 1024 neighbours. The representations (1024 dimensions) are the output of the last self-attention layer of the trained Transformer-based language model [Vaswani et al., 2017, Baevski and Auli, 2018]. The importance of the retrieved artefact in the output is determined by the manually set hyperparameter $\lambda \in [0, 1]$, resulting in linear (convex) interpolation.

The symbolic working of the model is shown in the following set of equations which is adapted from the original paper (the aggregator is the softmax function; $\text{LM}^{rep.}(X_{<i})$ is the vector representation of the prefix $X_{<i}$ by the trained language model).

$$\texttt{Encoder:} \qquad k = \text{LM}^{rep.}(X_{<i})$$

$$\texttt{Knowledge base:} \qquad \mathcal{B} = \{(\texttt{Encoder}(X_{<i}), X_i) | X \in D_{\text{train}}, i < |X|\}$$

$$\texttt{Retriever:} \qquad S = \{(r, v) | (r, v) \in \mathcal{N}_{1024}^{L^2}(k)\}$$

$$\texttt{Aggregator:} \qquad p_\xi(\hat{X}_i) \propto \sum_{(r,v) \in S} \exp(-||r - k||_2 \cdot v)$$

$$\texttt{Model:} \qquad p_m = \lambda \cdot p_\xi + (1 - \lambda) \cdot \text{LM}(X_{<i})$$

The authors also showed that using the training data as a knowledge base outperforms using them for training. This approach was built on the work of Grave et al. [2017] which, however, does not use the state-of-the-art Transformer-based language model but an RNN-based one. Furthermore, they use the inner product (IP) for similarity instead of the $L^2$ distance.

**Continuous Cache.** The previous approach is a continuation of using local vocabulary cache for language modelling, as proposed by Grave et al. [2016]. The effect of specific history size is examined as well. The usage of cache can be interpreted as using a small local dynamic knowledge base that is being updated after every prediction. This is motivated by the fact that especially rare words tend to occur more probably than by overall uniform distribution, given that they appeared

---

4. The negative $L^2$ distance is used instead of the common inner product as the similarity measure. This was shown empirically by the authors to work better.

in recent history. The LSTM [Hochreiter and Schmidhuber, 1997] hidden state size is again 1024-dimensional, although this time, it is not used for retrieval but only by the aggregator. For the fusion, the authors propose two methods: (1) linear interpolation, as seen in the previous language model and (2) joint softmax over the artefact and the language model output distribution.

The simplest of the two proposed models (linear combination) is symbolically described in the following equations (the aggregator is the softmax function; $\theta$ is a hyperparameter of the cache distribution).

$$\text{Encoder:} \quad k = \text{LSTM}^{hid.}(X_{<i})$$

$$\text{Retriever:} \quad S = \bigcup_{n<N} B_{i-n}$$

$$\text{Aggregator:} \quad p_\xi(\hat{X}_i) \propto \sum_{(r,v)\in S} \mathbb{1}_{v=\hat{X}_i} \exp(\theta \cdot k^T r)$$

$$\text{Model:} \quad p_m = \lambda \cdot p_\xi + (1-\lambda) \cdot \text{LSTM}(X_{<i})$$

$$\text{Knowledge base update:} \quad \mathcal{B}_{i+1} = \{(k, \arg\max \hat{v})\}$$

**Dynamic Gating.** The previously described language models use very late fusion, which is controlled by hyperparameter $\lambda$. Yogatama et al. [2021] propose an approach in which the model itself determines this parameter (now a vector) dynamically based on the current sample. The knowledge base (called long-term memory) is constructed in the same way from the training data as in k-Nearest Neighbours LM. They also introduce short-term memory in the model, which is able to attend to extended local context. Another difference is using two different models for the encoder (vanilla transformer) and the language model itself. The keys are 512-dimensional vectors and the retriever uses inner product for nearest neighbour lookup.

The following set of equations, adapted from the original paper, describes the behaviour of the model with respect to the long-term (episodic) memory:

$$\text{Encoder:} \quad k = \text{Transformer}^{rep.}(X_{<i})$$

$$\text{Knowledge base:} \quad \mathcal{B} = \{(\text{Encoder}(X_{<i}), X_i) | X \in D_{\text{train}}, i < |X|\}$$

$$\text{Retriever:} \quad S = \mathcal{N}_4^{IP}(k)$$

$$\text{Aggregator:} \quad p_\xi(v') = \sum_{(r,v)\in S} \text{softmax}(v, k) \cdot v'$$

$$\text{Model:} \quad g = \sigma(w_g^T k)$$

$$z = (1-g) \odot p_\xi + g \odot \text{LM}(X_{<i})$$

$$p_m = \text{softmax}(z, W)$$

Since the artefact is processed slightly more by the model compared to the other systems which we describe, e.g. Grave et al. [2016], Khandelwal et al. [2019], we classify it as *late* fusion as opposed to *very late* fusion. This dynamic gating was also applied to the cache knowledge base structure by Merity et al. [2016].

**Knowledge Graph LM.** All of the previous models utilized train-time knowledge base creation with keys representing the contexts and values of the next word predictions. Logan et al. [2019] utilize knowledge graphs to specifically increase language modelling performance on named entities.

We only describe the simplified retrieval component and omit the details, such as entity rendering and aliases.

At every position (for every query), the model makes a decision as to what type the following token will be: (1) non-entity, (2) unrelated entity or (3) related entity. In the first case, the token is predicted by the standard language model. The other two, however, utilize the knowledge graph access.

Formally there are two knowledge bases used: a static knowledge graph $\mathcal{KG}$ and a local graph $\mathcal{KG}_{<i}$ containing already encountered entities and their relations in the prefix. The model uses an LSTM unit, the hidden state of which is split into three components $[h_x; h_p; h_r]$ used for: (1) the token type decision, (2) parent entity prediction and (3) relation prediction. When the model makes a decision to predict an unrelated entity, it is sampled by a simple projection to the entity embedding space:

$$\text{softmax}(v_e \cdot (h_p + h_r))$$

The predicted entity $e$ is then added together with its immediate neighbours to the local graph:

$$\mathcal{KG}_{<i+1} = \mathcal{KG}_{<i} \cup \{(e, r, x) | (e, r, x) \in \mathcal{KG}\}$$

In case of a related entity, the model first predicts the parent, then the relation and finally the entity itself. When there are multiple entities in the local graph matching the restriction of the parent and the relation, it is sampled at random. The following set of equations describes the model behaviour in case of a new entity.

$$
\begin{aligned}
\texttt{Encoder:} \quad && p_p(e_p) =&\ \text{softmax}(v_p \cdot h_p) \\
&& p_r(r) =&\ \text{softmax}(v_r \cdot h_p) \\
&& \text{constrained by } \exists e : & (e_p, r, e) \in \mathcal{KG}_{<i} \\
\texttt{Retriever:} \quad && p_e \underset{\text{sample}}{\sim}&\ \{(e_p, r, e) | (e_p, r, e) \in \mathcal{KG}_{<i}\} \\
\texttt{Model:} \quad && p_m =&\ \text{renderer}(p_e) \\
\texttt{Update knowledge base:} \quad && \mathcal{KG}_{<i+1} = \mathcal{KG}_{<i} \cup & \{(e, r, x) | (e, r, x) \in \mathcal{KG}\}
\end{aligned}
$$

The main motivation for this approach is factual correctness in language modelling. Furthermore, it grants a higher degree of explainability and also allows for tighter manipulation and control of the data the model is working with (changing an entity in a relation has a direct impact on the produced output).

## A.2 Question Answering

There has been a lot of progress in question answering systems. However, the task of long-form question answering was recently shown to be problematic even with state-of-the-art Transformer-based systems [Krishna et al., 2021]. Furthermore, it has been suggested that dense representations are inadequate when scaled up to large index sizes [Reimers and Gurevych, 2020]. Despite that, the current trend is based on this retrieval mechanism [Lee et al., 2019, Guu et al., 2020, Lewis et al., 2020].

**Dense Passage Retrieval.**     We focus on DPR [Karpukhin et al., 2020] which describes a prototypical question-answering system built with knowledge base access and dense document embeddings.

DPR uses two BERT models to compute the document and question embeddings at the `[CLS]` token (768-dimensional). They are then fine-tuned so that the inner product (or $L^2$ distance or cosine similarity) of these two vectors is a good measure for document relevancy. The retrieved passages are reranked by a combination of the original similarity and the BM25 model [Robertson et al., 1995]. For every retrieved passage, the probability of a span (up to fixed maximum length) being selected is computed as the product of two tokens (representation from another BERT model) being the start and end ones, respectively. Reranking of the answers is done implicitly by choosing the span with the highest probability across all spans. The maximum similarity search is approximated to make it computationally feasible using FAISS [Johnson et al., 2019].

$$
\begin{aligned}
\text{Encoder:} \quad & k = \text{BERT}_{\text{query}}(q) \\
\text{Knowledge base:} \quad & \mathcal{B} = \{\text{BERT}_{\text{doc}}(d) | d \in \mathcal{KB}\} \\
\text{Retriever:} \quad & C = \arg\max_{d \in B}^{n} \text{sim}(q, d) \\
& C' = \text{Reranker}(C) \\
& \text{score given by: } \text{BM25}(q, d) + \lambda \cdot \text{sim}(q, d) \\
\text{Model:} \quad & P_{\text{start},i}(s) = \text{softmax}(C'_i w_{\text{start}})_s \\
& P_{\text{end},i}(t) = \text{softmax}(C'_i w_{\text{end}})_t \\
& P_{\text{selection}}(i) = \text{softmax}(C' w_{\text{selection}})_i
\end{aligned}
$$

For every passage, $P_{\text{selection}}(i)$ is the score that this passage contains the answer and for every span $P_{\text{start},i}(s) \cdot P_{\text{end},i}(t)$ is the score of a single span ($s$ to $t$) in a passage. In this specific model, the aggregator simply passes on all the retrieved passages, though some models simply concatenate all passages and pass it to the model as single string input [Izacard and Grave, 2020].

**Nearest Neighbour QA**     The reliance on training data is taken to its extremes by one of the models in Lewis et al. [2021] in their study of overlap (in terms of paraphrases) between test and train datasets for question answering. In their approach, they simply use nearest neighbours to retrieve the closest paraphrase to the query (using vector space model) to answer the question. The encoding is done either by the pretrained DPR retriever [Karpukhin et al., 2020] or by TF-IDF. As a consequence model can easily answer a question granted that the paraphrase is present in the training set.

$$
\begin{aligned}
\text{Encoder:} \quad & k = \text{DPR}_{\text{query}}(q) \\
\text{Knowledge base:} \quad & \mathcal{B} = \{(\text{DPR}_{\text{query}}(d) | d \in \mathcal{KB}\} \\
\text{Retriever:} \quad & o = \xi = \arg\max_{d \in B} \langle q, d \rangle
\end{aligned}
$$

Although this nearest neighbour model is a very specific corner-case of the artefact retrieval architecture, it can still be accommodated.

**CBR-KBQA**     Models utilizing case-based reasoning methods [Aamodt and Plaza, 1994] first retrieve similar cases which are then used in synthesising the current answer. For question answering,

such an architecture has been proposed by Das et al. [2021]. The retrieved similar queries also contain their logical forms (e.g. SQL or a graph query), based on which a logical form for the current query is constructed using a Transformer-based model BigBird [Zaheer et al.]. This logical form is then executed against a symbolic knowledge base and further refined by another component. The refinement step solves the issue of sparse knowledge bases and aligns each logical query edge This is performed either by pre-trained KB embeddings [Bordes et al., 2013] and similarity search or by using surface form similarity of edge names.

```
Encoder:                                              k = ROBERTA_BASE(q)
Symbolic knowledge base:         B_S = {(ROBERTA_BASE(d), f)|(d, f) ∈ KB_S}
Retriever:                                         C = arg max ⟨q, d⟩
Aggregator:       ξ = BIGBIRD(q[SEP]C_1^q[SEP]C_1^f[SEP]...[SEP]C_n^q[SEP]C_n^f)
                                                     ξ' = ALIGN(ξ, KB)
Retriever:                                         o = Execute(ξ', KB)
```

$$k = \textsc{Roberta}_{\textsc{BASE}}(q)$$
$$\mathcal{B}_\mathcal{S} = \{(\textsc{Roberta}_{\textsc{BASE}}(d), f)|(d, f) \in \mathcal{KB}_S\}$$
$$C = \arg\max_{d \in B}^{n} \langle q, d \rangle$$
$$\xi = \textsc{BigBird}(q\,\texttt{[SEP]}\,C_1^q\,\texttt{[SEP]}\,C_1^f\,\texttt{[SEP]}\,\dots\,\texttt{[SEP]}\,C_n^q\,\texttt{[SEP]}\,C_n^f)$$
$$\xi' = \textsc{Align}(\xi, \mathcal{KB})$$
$$o = \texttt{Execute}(\xi', \mathcal{KB})$$

Advantages of this approach include a higher degree of explainability, higher performance on complex compositional questions and, because the model is non-parametric, extendability of the set of schemas and the knowledge base. It also demonstrates how some architectures may use more the first retrieval to craft a key for a second retrieval.

**PullNet**   The usage of multiple knowledge base sources is further developed by Sun et al. [2019b]. In the proposed model, a subgraph is iteratively constructed. The subgraph starts with containing only the query and in every iteration step, dubbed *pull*, a node is expanded. The transition from text to a knowledge graph structure is performed using an entity linker [Ji et al., 2014]. The output of this process is a set of triples (subject, predicate, object). Given query $q$, a textual knowledge base $\mathcal{KB}$ and a knowledge-graph $\mathcal{KG}$, the high-level simplified working of PullNet can be summarized as:

$$C_0 = \textsc{Entities}(q)$$
$$C_{i+1} = C_i \cup \{\textsc{PullDocs}(e, q, \mathcal{KB}) \cup \textsc{PullFacts}(e, q, \mathcal{KG})|v \in \textsc{PullNodes}(C_i)\}$$
$$o = \textsc{ClassifyAnswer}(C_T)$$

(Retriever / Model labels:)
- Retriever: $C_0 = \textsc{Entities}(q)$
- Model: $o = \textsc{ClassifyAnswer}(C_T)$

The operation PULLDOCS retrieves the most relevant documents to query $q$ using IDF, given the constraints of the entity $e$ (documents are linked to entities). The operation PULLFACTS retrieves facts from the knowledge-graph given the constraint of the entity $e$ being the subject or the object. The ordering is based on the inner product between a hidden state LSTM representation of the query $q$ and an embedding of the fact relation.

In the context of the artefact retrieval architecture, this model's working can be encapsulated in a single retrieval component that performs multiple knowledge base accesses followed by the model performing classification on top of the resulting subgraph.

**Universal Schema**   A similar approach to joining the reasoning capacity of models over structured knowledge bases and the amount of information on the web has been studied by Das et al. [2017]. Universal Schema [Riedel et al., 2013], upon which this model is based, is a way to embed knowledge base data universally in a matrix form. The task for which this is used is called fill-in-the-blank

question answering and each answer is a single entity. Given query input $q$ and the memory $\mathcal{M}$ the computation is as follows:

$$\text{Encoder:} \qquad c_0 = \text{BiLSTM}(q)$$

$$\text{Retriever/Attention:} \qquad c_t = W_t\big(c_{t-1} + W_p \sum_{(k,v)\in\mathcal{M}} (c_{t-1} \cdot k)\,v\big)$$

$$\text{Model} \qquad o = \arg\max_{e_i\in\mathcal{E}} c_t \cdot e_i$$

Due to the use of memory networks [Weston et al., 2014] and iterative attention, the distinction between model computation and retrieval becomes blurred. The intermediate context $c_t$, created by attending over the memory can be considered an artefact.

### A.3 Fact Checking

Despite the research in using the masked language modelling capabilities of pre-trained models for fact checking [Lee et al., 2020], traditional methods rely on external knowledge base access to verify claims. This has the advantage of higher explainability, which is essential in the context of fact-checking. The pipelines then usually consist of evidence retrieval and verification which maps well to our proposed abstract artefact retrieval model.

The release of FEVER: a large-scale dataset for fact extraction and verification [Thorne et al., 2018] introduced a benchmark which was utilized by multiple works, such as by Nie et al. [2019] or Nadeem et al. [2019]. We describe the components of the latter.

**FAKTA** The query encoder of FAKTA filters out words that are not verbs, nouns or adjectives. And appends to it named entities from the claim. The retriever then fetches relevant documents to this query and if none are found, relaxes the query by incrementally omitting the last tokens. The model uses several sources of knowledge bases, some of them external such as Google, Bing and Yahoo). This step is followed up by a re-ranker based on important words (determined by their POS tags) from the document title and the claim. Relevant documents are further filtered by a CNN based on the work of Xu et al. [2019]. Another CNN network determines the document stance (agree, disagree or discuss). Given a claim $q$ the pipeline is as follows:

$$\text{Encoder:} \qquad k = \text{Filter}_{\text{N,V,ADJ}}(q) \frown \text{Filter}_{\text{named entity}}(q)$$

$$\text{Retriever:} \qquad C = \text{SEARCHENGINE}(k_{<i})$$

$$(\text{such that } i \text{ maximal and results non-empty})$$

$$C' = \text{RERANKER}(C)$$

$$C'' = \{d|\text{CNN}_{\text{relevancy}}(d) = \text{relevant})\}$$

$$\text{Model:} \qquad o = \{\text{CNN}_{\text{stance}}(d)|d \in C''\}$$

The output of the model is a set of relevant documents with labels regarding the claim. These inferences can be averaged into a single number which describes how likely it is that the claim is factually true.

### A.4 Knowledgable Open Dialogue

Finally, we focus on open dialogue with utterance generation dependent on a knowledge base. This improves the insufficiencies of parametric model memory in open-ended dialogue centred around

facts. It also increases the explainability of each utterance as it can be traced (with various scores) to the source in the knowledge base. We will focus on the task and model description by Dinan et al. [2018] though there have also been previous works by Dodge et al. [2015] or in the context of a commonsense knowledge base by Young et al. [2018], Wu et al. [2020].

**Wizards of Wikipedia**  In the pipeline proposed by Dinan et al. [2018], the pool of retrieved documents provided by the method of Chen et al. [2017] in the form of a term matrix and a simple inverted index lookup. This pool is fixed so that the results are comparable to their experiments with human annotators. This part could however also be automated and replaced by a more advanced model which could be finetuned. The artefact in this pipeline is the weighted average of retrieved documents based on the dot product between the encoded representation and the encoded topic.

Given the conversation history $(q_1, q_2, \ldots, q_n)$ and topic $t$ the first model proposed by Dinan et al. [2018] can be described as:

$$\texttt{Encoder:} \qquad k = \bigcup_{x \in \{q_{n-1}, q_n, t\}} \textsc{TermVector}(x)$$

$$\texttt{Retriever:} \qquad C = \textsc{InvertedIndexLookup}_7(k, \mathcal{KB})$$

$$C' = \{\textsc{TransformerEnc}(\text{Title}(d) \frown \text{Paragraphs}(d)_1) | d \in C\}$$

$$\texttt{Aggregator} \qquad \xi = \sum_{d' \in C'} d \times (d \cdot \textsc{TransformerEnc}(t))$$

$$\texttt{Model:} \qquad o = \textsc{TransformerDec}(\textsc{TransformerEnc}(q) + a)$$

The next utterance knowledge source is then dependent on the last two utterances and the topic. The artefact is then added to the context embedding and similarly to the description in Figure 2, it simply moves the current vector in the embedding space, hopefully, in the right direction. This can be considered intermediate fusion since the artefact has yet to be processed by a Transformer model.

