# OpenReview forum: "Artefact Retrieval: Overview of NLP Models with Knowledge Base Access"
_AKBC.ws/2021/Workshop/CSKB — CSKB_

### Official Review · Reviewer_o61a · 2021-09-16

**Rating:** 7
**Confidence:** 3

**Review:**

This paper aims to provide a formalism for NLP models that retrieve knowledge from an external nonparametric memory and how the retrieved knowledge is fused into the model. The hope is that a formal exposition would enable us to uncover combination of mechanisms that have not been tried for a given task (and might have been used for other task) or a combination of mechanism which has not been tried at all.

The paper starts with a general description of retrieval based model which retrieves knowledge based on a query vector which is obtained by encoding the task input. The retrieved knowledge is then aggregated by an aggregator to produce an artefact which is used as an input to the model to generate an output for the task. An artefact is defined as an intermediate object that depends on the elements retrieved from the knowledge base and can be used to improve task performance during training and inference.

The paper also lays down a general description of how the knowledge source is constructed. They can be structured knowledge graphs containing knowledge triples, unstructured document collection (e.g. for open-domain QA) and also the training set (e.g. for models which retrieve KNN training examples during inference)

Strengths:

1. The paper lays out a general framework for knowledge retrieval models for various NLP tasks. The abstraction is quite general, and it covers quite a few models for various talks and is an interesting read.

Weakness:

1. In sec 3.2, I think it will be concatenating x and "a", instead of "y". On another note, I think artefact should be denoted by \kappa (or k) as denoting it with "a" can confuse the reader as "a" is often used to denote answer in QA tasks.
2. Other examples of early fusion models worth mentioning are Das et al 2017 and Sun et al 2019.
3. I find it a little hard to understand why in figure 2, early and late fusion models are described in terms of "loss".  Loss is something which is usually computed during training and there is no guarantee that the loss of a model with artefact would be lower than the loss of the same model without artefact. It might well be the case that a model with extra knowledge might take more time to train and have higher loss than a model without artefact, but actually converge to a point where the model is able to generalize better. I think the explanation should be modified to include task performance at inference (and I think that is the intended interpretation of the distance measure between the output and the projection) rather than training loss.
4. I like the general description of KB source and key and value type in Sec 3.4 and 3.5. The paper could include another class of work that retrieves other similar question, answer pair in the training set to answer a given question. For example, in Lewis et al 2021, a model is proposed that retrieves paraphrases of the question from the training set and returns the answer. This model can therefore easily answer a question if the paraphrase is present in the training set. More recently, Das et al 2021 proposed a model based on case-based reasoning, in which they retrieve other similar questions and logical forms (SQL queries) and uses them to generate logical forms for new query. Since they use both query and logical forms, they can answer questions even if the exact paraphrase is not present in the training set.
5. Minor: The citation of KILT says Fan et al 2020, but I think it should be Petroni et al 2020. Might be an error in the bib file?

Missing References:

Das et al 2017: Question answering on knowledge bases and text using universal schema and memory networks ACL 2017

Sun et al 2019 PullNet: Open Domain Question Answering with Iterative Retrieval on Knowledge Bases and Text EMNLP 2019

Lewis et al 2021: Question and Answer Test-Train Overlap in Open-Domain Question Answering Datasets EACL 2021

Das et al 2021: Case-based Reasoning for Natural Language Queries over Knowledge Bases EMNLP 2021

---

> ### Author Response · Authors · 2021-09-30
> **Review Feedback**
>
> Thank you for your detailed comments and suggestions.
>
> 1. We opted for $\xi$ (xi) for denoting the artefact object because $\kappa$ (kappa) could conflict with the denotation of a knowledge base (though in our case we use $\mathcal{KB}$).
> 2. Case-based reasoning QA (Das et. al. 2021) is highly interesting in this context, particularly because it uses a fusion mechanism different from the other described models (creation of an executable query). For completeness, we also include nearest-neighbour QA (Lewis et. al. 2021) though the pipeline is fairly simple and does not benefit much from being explained in the context of artefact retrieval.
> 3. The term "loss" was used to describe distances between points in intermediate projections. A key remark is that it is the distance to the _closest_ point in the space which results in the correct prediction. We changed the paragraph to make it more explicit that we are talking about computations during inference.
> 4. PullNet (Sun et. al. 2019) and Universal Schema (Das et. al. 2017) make a good addition to the list of examples because of their iterative nature in knowledge base / memory access.

---

### Decision · Program_Chairs · 2021-09-18

Accept